



# Lessons learnt from analysing the causes of casualties in the Sichuan Changning $M_s$ 6.0 earthquake

Wei Wang[1,2], Hong Chen[1], Xiaolin Jiang[3], Lisiwen Ma[1], Minhao Qu[4]

[1]Institute of Crustal Dynamics, China Earthquake Administration, Beijing 100085, China
[2]Institute of Engineering Mechanics, China Earthquake Administration, Harbin 150080, China
[3]Sichuan Earthquake Administration, Chengdu 610041, China
[4]National Earthquake Response Support Service, Beijing 100049, China

*Correspondence to*: Hong Chen (chenhongicd@163.com)

**Abstract.** This paper summarizes the $M_s$ 6.0 earthquake disaster that occurred in Changning, Sichuan Province, China, on 17
June 2019. Additionally, a description of the disaster emergency response is provided. As determined by on-site investigation and analysis, the direct causes of earthquake casualties are summarized. Among them, 66% of the casualties can be attributed to substandard seismic protection, 25% to improper earthquake risk avoidance, 6% to earthquake-induced geological disasters, and 3% to pre-existing diseases exacerbated by the earthquake. Responding to the causes of these four casualty types, we summarize 4 lessons, which are to build in accordance with seismic protection requirements; strengthen publicity and
education awareness of earthquake emergency evacuation and training for the population; investigate and evaluate potential geological disaster sources in the stricken area, along with conducting prevention and control; establish cooperation with health organizations, focusing on people who are older or have serious illnesses and conducting earthquake evacuation training and psychological counselling.

## 1 Disaster characteristics and disaster situation

An $M_s$ 6.0 ($M_w$ 5.8) earthquake struck Changning County, Yibin City, Sichuan Province, China, at 22:55:43 on 17 June 2019 (14:55:43 UTC on 17 June 2019). The epicentre was located at 28.34°N, 104.90°E with a focal depth of 16 km (China Earthquake Networks Center (CENC), 2019). This earthquake was located in the southeastern part of Sichuan Province and occurred on secondary faults near the Changning anticline structure close to the edge of the Sichuan Basin. It developed on a different structure from the Wenchuan $M_s$ 8.0 earthquake and the Lushan $M_s$ 7.0 earthquake, which occurred in the
Longmenshan fault zone. The aftershocks of the earthquake remained active, and the results of the focal mechanism solution showed a main thrust tectonic quake (Hu et al., 2019). After the earthquake, 62 incidents of $M_s$ 2.0 and higher aftershocks were recorded by 12:00 on 4 July 2019. Among them were 4 earthquakes of magnitude 5.0 to 5.9, 6 earthquakes of magnitude 4.0 to 4.9, and 52 earthquakes of magnitude 3.0 to 3.9 (Earthquake Administration of Sichuan Province, 2019a).

The population density of Changning County was 340 people per km$^2$, according to the statistics from the Yibin City Health
Committee. The earthquake killed 13 people, 7 people were critically injured, and 14 people were seriously injured. The local



medical and health organization made rounds of visits to treat more than 6,200 wounded people (Earthquake Administration of Sichuan Province, 2019b). More than 50,000 houses collapsed and/or were seriously damaged. According to preliminary statistics, the direct economic loss was approximately 8.889 billion yuan (China National Radio (CNR), 2019a).

The earthquake disrupted 1207 stations and 43 power supply lines in the power supply networks of Changning County, Gong County and Xingwen County and resulted in power outages for 71,600 users, which were fully repaired by 21:00 on 19 June. The Yixu Expressway near the earthquake zone, the four toll stations in Yibin territory on the Yilu Expressway, and some of the ramps to the Naqian Expressway were closed due to the earthquake. On the left side of S309 Gugao Road K234+550 (Dongdi Town, Changning County), a mountain collapsed, causing road obstruction and reducing the effective roadway width by half. There were 52 Yibin base station interruptions and 14 optical line terminations, but communications were not

interrupted. The disrupted base stations in Gong County and Changning County accounted for 16.49%, affecting approximately 1,600 broadband users (People's Posts and Telecommunications News (PPTN), 2019).

The Earthquake Administration of Sichuan Province published the earthquake intensity distribution and the earthquake intensity map of the Sichuan Changning $M_s$ 6.0 earthquake on 20 June based on an investigation of the damage to structures and the distribution of casualties, as shown in Fig. 1.

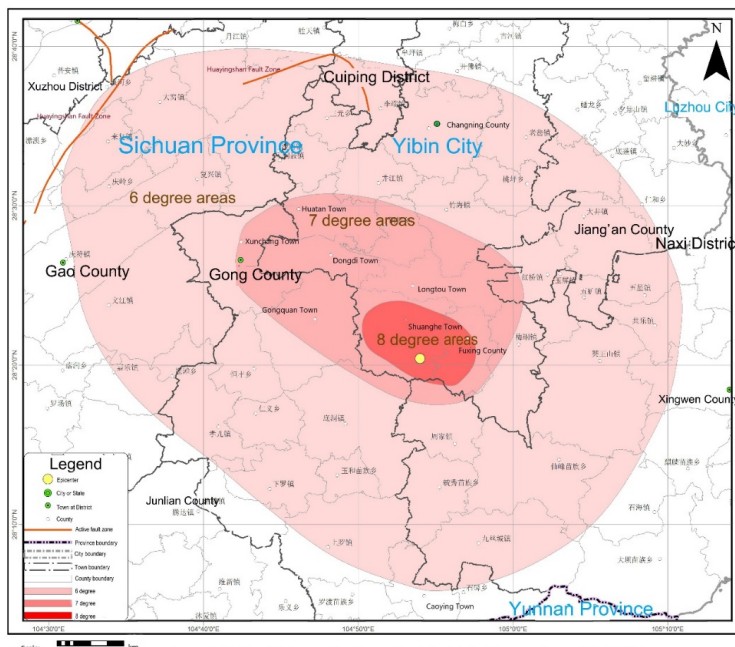


**Figure 1: The earthquake intensity map of the Sichuan Changning $M_s$ 6.0 earthquake ©EASP (Earthquake Administration of Sichuan Province, 2019c).**





The highest intensity of the earthquake was degree Ⅷ (degree 8). The total area of the degree Ⅵ (degree 6) area and above was 3058 square kilometres. The long axis of the isoseismic line was northwestward, with a long axis of 72 kilometres and a

short axis of 54 kilometres. The earthquake affected mainly Changning County, Gao County, Gong County, Xingwen County, Jiang'an County and Cuiping District of Yibin City. Among them, the degree Ⅷ (degree 8) area occupied 84 square kilometres, mainly involving 3 townships. The degree Ⅶ (degree 7) area occupied 436 square kilometres, mainly involving 14 townships. The degree Ⅵ (degree 6) area occupied 2538 square kilometres, mainly involving 58 townships. In addition, some areas outside the degree Ⅵ area were affected, and some older houses were damaged. (Degree Ⅷ area: a few houses with masonry-

timber structure were destroyed, and most of them were seriously damaged; houses with brick-concrete structure were destroyed individually, and most of them were moderately damaged; a few houses with frame structure were severely damaged, and most of them were slightly damaged. Degree Ⅶ area: houses with masonry-timber structure were destroyed individually, and most of them were moderately damaged; a small number of houses with brick-concrete structure were slightly damaged, and the houses with frame structure were slightly damaged. Degree Ⅵ area: a few houses with masonry-timber structure were

destroyed; several houses with brick-concrete structure were slightly damaged.)

## 2 Emergency response

After the earthquake, the Ministry of Emergency Management (MEM) of PRC immediately launched an emergency response. Through the command centre, the MEM, the China Earthquake Network Center, the Department of Emergency Management of Sichuan Province, the Sichuan Provincial Fire Rescue Corps, and the earthquake rescue site initiated video connections to

assess the disaster situation and to deploy rescue and relief operations. Sichuan Province launched a grade II emergency response and established provincial "6·17" earthquake relief and emergency rescue joint headquarters to guide and deploy relief and rescue operations. The entire emergency response period of this earthquake lasted 10 days and ended at 12:00 on 27 June 2019.

Sichuan Province is an earthquake-prone area. Especially after the earthquakes in Wenchuan (2008, $M_s$ 8.0), Lushan (2013,

$M_s$ 7.0) and Jiuzhaigou (2017, $M_s$ 7.0), Sichuan Province has accumulated rich experience in earthquake emergency response and disposal and has a complete earthquake emergency plan system. After this earthquake, a multi-sector joint response was launched immediately from the central government to work with local people and various departments. The MEM and the National Health Commission sent a joint working group to the disaster area to guide and coordinate relief and rescue operations. The Ministry of Natural Resources, the Ministry of Water Resources and other relevant departments instructed the local

authorities to investigate the potential risks and dangers around the earthquake zone. Bridge experts, geologists and rescue teams, which were organized by the Sichuan and Yibin municipal governments, provided relief efforts. Emergency relief tents, quilts, folding beds and other disaster relief materials were delivered to the disaster area. Lifeline project repairs and emergency rescues were carried out in various departments of the disaster area and across institutions, as shown in Table 1.


**Table 1: Lifeline project repairs (China National Radio (CNR), 2019b) and the emergency response by rescue forces and social**
**organizations.**

| Sectors | Responding departments and organizations | Specific aspects of emergency response work |
|---|---|---|
| Transportation | The Department of Transportation of Sichuan Province launched a grade II response to repair the roads immediately. | The Department of Transportation of Sichuan Province sent working groups and experts to the earthquake site and quickly deployed emergency rescue materials and steel bridges in addition to emergency personnel to assemble and await orders. The local transportation department of Yibin organized the preliminary emergency reconstruction work and sent personnel to the affected areas to investigate the disaster.<br>On the Yixu Expressway, to address dangerous conditions, the working group excavated more than 400 cubic metres, repaired more than 30 points by crack pouring, and treated the pavement over approximately 2000 metres.<br>The working group eliminated the danger on provincial road S443 by repairing the road within 7 days after the earthquake, ultimately disposing of 10,000 cubic metres of earth and stone.<br>Damaged rural roads and county township roads were repaired within 4 days after the earthquake.<br>Eight passenger lines were rendered non-operational due to the earthquake. They were fully restored within 4 days after the earthquake, and the passenger traffic resumed regular operation. |
| Electric power | The Sichuan Electric Power Company of the State Grid Corporation of China immediately launched an earthquake disaster grade II response. | Within three hours after the earthquake, an emergency power generation vehicle provided power supply to the epicentre refuge and rescue forces.<br>Within five hours after the earthquake, emergency lighting was provided for emergency shelters and resettlement sites in the four severely affected areas.<br>Within two days after the earthquake, the power was fully recovered; 860 people and 190 vehicles participated in restoring service. |
| Communication | Sichuan Mobile, Sichuan Telecom, Sichuan Unicom and Sichuan Tower Corporation all launched urgent earthquake emergency responses. | Within 4 days after the earthquake, the three major operators addressed a total of 294 communication base stations, all of which were repaired, and communications in the earthquake-stricken areas returned to normal. |
| Natural gas | Natural gas company | More than 50,000 household gas users in the earthquake-stricken areas were shut down, and the gas line safety was checked, with 231 potential dangers detected. Within 4 days after the earthquake, all potential dangers were rectified, and the gas supply was restored. |
| Government rescue force (The People's Government of Sichuan Province, 2019) | Deployed immediately | As of 9.5 hours after the earthquake and by 8:20 on 18 June, all 14 people trapped in collapsed buildings were rescued by the rescue teams of the Fire Fighters, People's Liberation Army, Armed Police, and the National Mine Emergency Rescue Team (Sina.com.cn, 2019).<br>The People's Liberation Army and the Armed Police:<br>The Sichuan Provincial Military Region and the Sichuan Provincial Armed Police Corps immediately launched an emergency plan and dispatched a task force to transport professional rescue equipment for prompt deployment to the disaster area.<br>A total of 2,642 officers and enlisted soldiers of the People's Liberation Army and Armed Police Forces and militia participated |



| | | |
|---|---|---|
| | | in first-line rescue, disaster investigation and warning duty in the disaster area. A total of 40 people were rescued, 623 people were evacuated, and 1,600 people affected by the disaster were transferred. In addition, 565 sets of tents were built, 81 tons of materials were delivered, 4.6 kilometres of roads were drained, and more than 30 dangerous regions were made safe. <br> Fire Fighters: <br> After the earthquake, 21 members of the Yibin Fire and Rescue Task Force immediately deployed to the epicentre in Shuanghe Town to assess the disaster. The Yibin fire brigade earthquake rescue force sent 10 cars and 42 members to lead the team to the epicentre in Shuanghe Town. The provincial fire rescue team's full-time command and the surrounding six fire rescue detachments dispatched 63 fire engines and 302 firefighters to the disaster relief site to carry out comprehensive investigation and rescue work. Within 2.5 hours after the earthquake, the Yibin Fire and Rescue Detachment successfully rescued three trapped people. <br> The National Mine Emergency Rescue Team: <br> The National Mine Emergency Rescue Furong Team sent three rescue vehicles, and 30 team members arrived at the disaster relief site and rescued an injured person with earthquake relief-related equipment and medical emergency equipment before 2:00. <br> Traffic Police: <br> A first-responder crew of 23 people from the Yibin City Traffic Police Detachment and 83 people from the Changning County Traffic Police Detachment responded to the Shuanghe site to inspect and control the road surfaces. |
| Social organizations (Emergency Rescue Center of MEM, 2019) | China Red Cross Society launched a grade III emergency response. | The Red Cross Society of China dispatched a disaster relief working group to the disaster area to evaluate the disaster situation, assess the on-site needs, guide the local Red Cross to carry out disaster relief work, and distribute 100 tents, 2000 family emergency kits, 2000 quilts and 1000 jackets from the Chengdu Disaster Preparedness and Disaster Relief Center. |
| | Eighty-nine social organizations | According to statistics from 25 June, 89 social organizations were involved and participated in the Changning earthquake response. Among them, a total of 18 rescue teams, 15 social service agencies, 5 professional service organizations, and 10 foundations arrived at the disaster relief site, for a total of 48 organizations. Two foundations did not visit the site but provided funds or material support off-site; two professional service organizations provided online services; and 37 organizations organized relief online. |

## 3 Investigation and analysis of the causes of casualties

The earthquake caused a total of 13 deaths, 7 critical injuries and 14 serious injuries, or 34 casualties in all. The on-site investigation of the assessment team of the Sichuan Earthquake Administration collected information on 32 casualties (omitting one critically injured person and one seriously injured person).




**3.1 Distribution**

Among the casualties, 23 people were located in the degree VII and VIII areas of earthquake intensity. Nine people were located in the degree VI area of earthquake intensity. No reports of death, critical injuries or serious injury were found in the areas of degree V or below, as shown in Fig. 2. Casualties were characterized by a concentration in high-intensity areas, consistent with the intensity division.

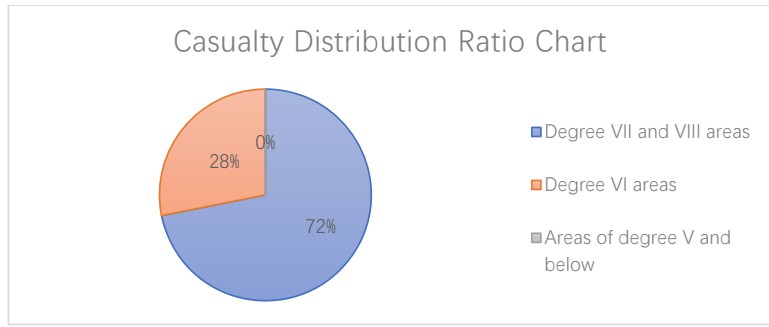

**Figure 2: Casualty distribution ratio chart.**

**3.2 Age**

The oldest of the 32 casualties was 92 years old, and the youngest was only 3 months. The casualties comprised 1 child, 3 teenagers, 6 young people, 13 middle-aged people and 9 elderly people, as shown in Fig. 3. The proportion of young people and middle-aged people was more than half, and the proportion of elderly people was less than 30%. These rates do not reflect an age advantage in the process of evacuation. The reason may be that the earthquake occurred at midnight; all the casualties were indoors and may have fallen asleep, and there was almost no time to perform the necessary evacuation.

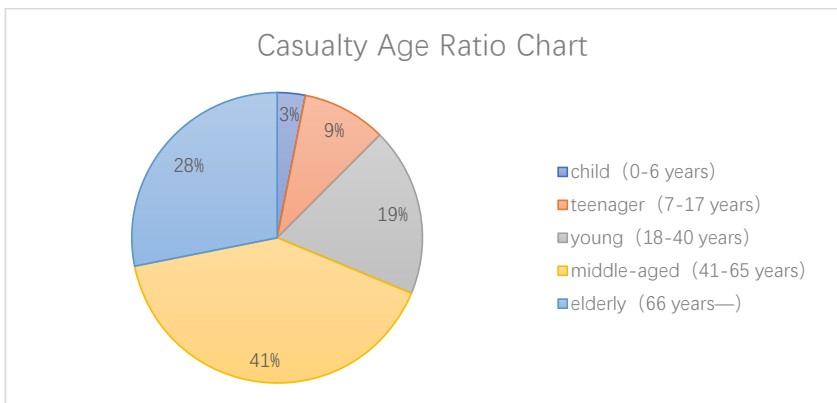

**Figure 3: Casualty age ratio chart.**





**3.3 Causes**

The causes of injury and death of 32 persons can be classified into four categories, as shown in Fig. 4. The first cause is that the structure of the house was seriously damaged or collapsed, and the debris caused injury or death to 21 people in total. Second, self-built houses were located on the unstable side of the slope, and the earthquake caused boulders to roll down and destroy the houses, affecting 2 people. Third, after the earthquake, people fled and escaped in a panicked state. Due to improper

evacuation measures, 8 people were injured (there was no death in this case). Fourth, one victim suffered from cardiovascular and cerebrovascular diseases and succumbed to cerebral haemorrhage due to mental stress and excessive physical exertion during the process of evacuation transfer.

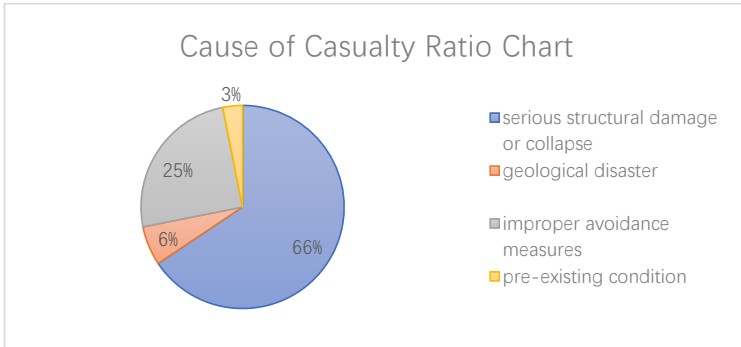

**Figure 4: Cause of casualty ratio chart.**

Among the 13 deaths, serious damage or collapse of the house structure resulted in 10 deaths, 2 people were killed by rockfall, and 1 victim died of illness, as shown in Fig. 5.

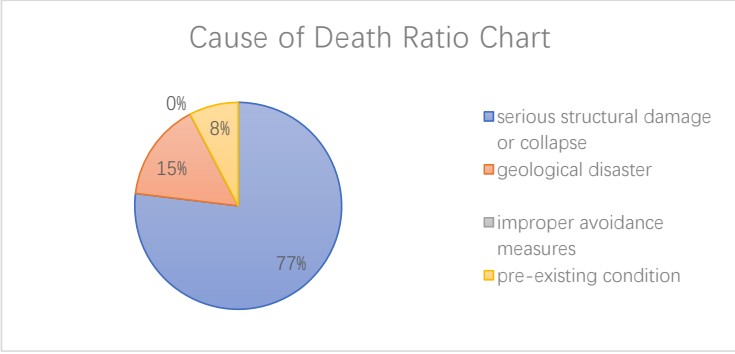

**Figure 5: Cause of death ratio chart.**





Among the 19 critically injured and seriously injured patients, 11 were injured due to serious damage or collapse of the house

structure, and 8 were injured due to improper evacuation measures, as shown in Fig. 6.

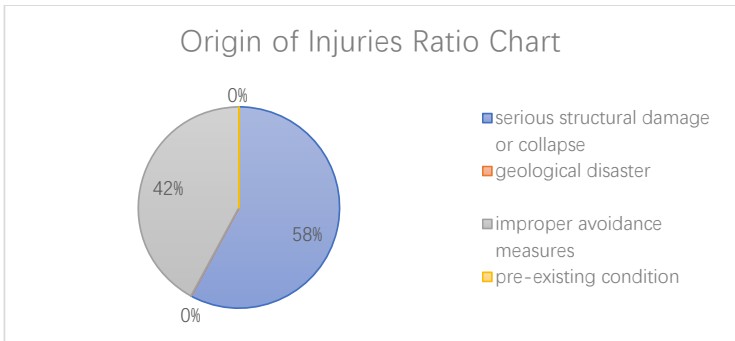

**Figure 6: Origin of injuries ratio chart.**

**3.4 Analysis**

Among the 13 deaths, one person in Gong County was killed by the collapse of a house that was self-built with prefabricated

panels, as shown in Fig. 7. In this village, there remained a large number of self-built prefabricated houses without constructional columns, and their anti-seismic performance was poor. All of these houses were damaged at different levels in this earthquake.

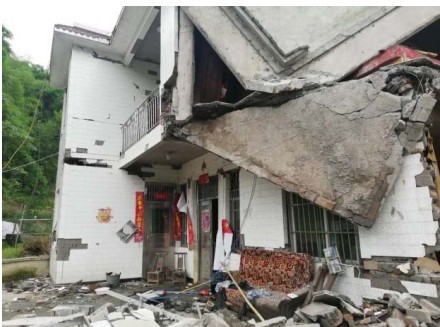

**Figure 7: Victim's house in Yuchi village, Gongquan town, Gong county.**

The deaths of 9 people in Shuanghe Town were caused by the serious destruction or complete destruction of the house structures, and the people were buried under pieces of debris. The houses in which some of the deceased had lived were houses with masonry-timber structure aged more than 50 years; these houses had almost no ability to resist medium-sized earthquakes, as shown in Fig. 8.



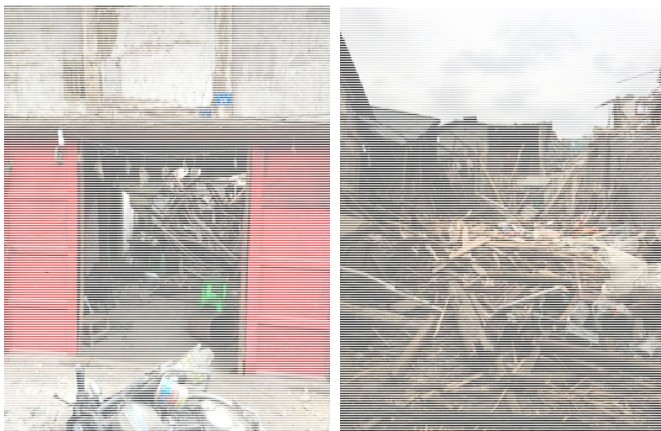

**Figure 8: Shuanghe town, where some of the Masonry-timber structures collapsed or were completely destroyed.**

Two people died in Xunchang Town, Gong County, because the self-built houses were located on the unstable side of the slope. The earthquake caused a rockfall that destroyed the houses, as shown in Fig. 9.

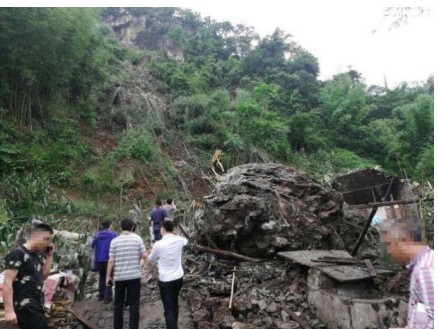

**Figure 9: Xunchang town, Gong county: rockfall and a victim's house.**

One person died in Caoying Town, Gong County, due to causes indirectly related to the earthquake. He suffered from cardiovascular and cerebrovascular diseases and succumbed to cerebral haemorrhage due to mental stress and excessive physical exertion during the evacuation transfer.

The causes of critical injuries and serious injuries in this earthquake are typical and can be summarized into the following two categories:

The self-built houses lacked anti-seismic design, and their ability to resist destructive earthquakes was poor. Some of the house structures were seriously damaged or completely destroyed. Eleven people in total were injured by pieces of debris.




People lacked scientific cognition of earthquake disasters or were unfamiliar with specific measures for reasonable evacuation, causing them to run in a dimly lit environment at night or return to the unsafe interior of a house to retrieve their property, resulting in 8 injuries in total.

In summary, the main cause of casualties in this earthquake was that the seismic protection of houses did not meet the standards, the structures of houses were seriously damaged or completely destroyed, and the collapse of houses caused approximately 66% of casualties. The residences of the casualties were self-built houses without structural support and lacked anti-seismic design.

Injuries caused by inappropriate earthquake evacuation also accounted for approximately 25% of the casualties. After the

earthquake, fearful and panicked escapes under emergency conditions were predominant, and reasonable evacuation measures were seldom taken.

The casualties caused by a geological disaster (rockfall) due to the earthquake resulted in two deaths, with a proportion of 6%. The house locations were not properly planned, and the geological disaster caused by the earthquake directly contributed to the casualties.

One person (representing 3% of the casualties) did not die directly due to the earthquake. Instead, he succumbed to cerebral haemorrhage due to disease during the process of evacuation transfer and died receiving medical treatment.

**4 Lessons learnt**

Sichuan Province is an earthquake-prone area. Especially after the earthquakes in Wenchuan, Lushan and Jiuzhaigou, Sichuan Province has utilized its rich experience in earthquake emergency response and disposal to develop a complete earthquake

emergency plan system. After this particular earthquake, multi-sector joint responses were launched immediately from the central government to involve local people and various departments. Lifeline systems such as transportation and electric power were quickly restored after the earthquake. Various rescue forces were dispatched quickly, and all of the trapped people were rescued.

During the Changning earthquake, in addition to satellite remote sensing, UAV telemetering, big data by smartphones and

other new technologies and high-tech means used in earthquake rescue, Sichuan Provincial People's Hospital and China Mobile quickly responded with cross-border cooperation, and 5G technology was applied to earthquake relief medical rescue for the first time in the world. An ambulance equipped with 5G technology went to the disaster area to realize the integration of remote diagnosis, trans-shipment and treatment and greatly shortened the rescue response time (Science and Technology Department of Sichuan Province, 2019).

By analysing the causes of casualties, the following lessons were learnt:

1. The structures of houses were severely damaged or completely destroyed, and the collapses of houses caused 66% of the casualties in the earthquake. The residences associated with the casualties were self-built houses without structural support and lacked anti-seismic design. A large number of buildings in remote areas of China (especially in



mountainous areas and rural areas) have not applied seismic protection measures. For construction of buildings in the important seismic monitoring and prevention regions of China, governments at all levels should encourage people to build or reinforce existing houses in accordance with seismic protection requirements through incentives such as policies and insurance.

2. In addition, 25% of the casualties in this earthquake were caused by improper evacuation. For example, a critically ill patient in the earthquake was safely moved outside after the earthquake but returned to the room to retrieve personal belongings. The house, self-built with prefabricated panels, collapsed and severely injured this person. The local government should strengthen publicity and education awareness about earthquake emergency evacuation and provide training to cover all ages.

3. Sichuan's geological structure is complex, earthquakes and fault activity are frequent, and the terrain is highly disarrayed. Earthquakes often cause landslides. Although earthquake geological disasters were not the main cause of death in this earthquake, based on past experience, such disasters are still an important cause of casualties. Governments at all levels should carry out investigations of potential seismic geological hazards, conduct hazard and threat assessments based on survey data, and identify key sources of potential geological hazards based on the results of the assessment. First, it is necessary to scientifically and reasonably plan the site selection of buildings, enforce spacing from the sources of potential geological disasters and estimate the extent of damage that may be inflicted. Second, effective measures should be taken to comprehensively control potential geological disasters to prevent casualties caused by earthquakes.

4. One of the deaths in this earthquake was not caused directly by the earthquake. Instead, the patient died in the process of evacuation transfer because of a cerebral haemorrhage due to a previously existing illness, and he died receiving medical treatment. Local governments should cooperate with health organizations to focus on people who are older or have serious diseases. The governments should provide earthquake evacuation training and psychological counselling to citizens and enable them to seek help from medical or related organizations in a timely manner during the process of transfer. Primarily, governments at all levels should guide citizens to scientifically understand earthquake disasters and avoid panic caused by sudden earthquakes. The second aim should be to guide citizens to focus on self-protection during the evacuation process to avoid life-threatening injuries.

**Code availability**

**Data availability**

Data used in this article have not been deposited to respect the privacy of users. The data can be provided to readers upon request.




**Competing interests**

None.

**Acknowledgements**

We thank the assessment team of the Sichuan Earthquake Administration for the on-site investigation that collected information on casualties.

**Funding**

This work was supported by the China Earthquake Administration [grant number 1840717266].

**Role of the Funding Source**

None.

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
