# Peer review of "Lessons learnt from analysing the causes of casualties in the Sichuan Changning $M_s$ 6.0 earthquake"

_Natural Hazards and Earth System Sciences, 2020_

## Referee Comment (RC1) · Alexandra Carvalho (Referee) · 27 May 2020

It is interesting and important to analyse the causes of casualties. Nevertheless, it is needed to keep in mind that the sample is small and probably not sufficient for a statistical analysis. Lessons learnt are known already, but even so they are well summarized. Probably should be interesting to compare results with other studied earthquakes?

1) Abstract

I would considered to write in the abstract the total of people (32) before referring that 66% ..... In that way it is possible to have an idea about the number we are talking about. If not, it seems sensationalist.

[Figure]

2) section 1. Disaster characteristics...

a) I would like to see a figure (Fig 1a or fig 1b) showing the tectonics and the location of the epicenters of the two or three earthquakes that are mentioned (two in section 1 and a third one at section 2).

b) lines 30-31: how many people has the Changing County and how many houses? (you mentioned 50000 houses collapsed and /or were seriously damaged. It should be interesting to have an idea of this representation). And can you add the total amount ( 8889 billions yuan) in dollars, as well?

3) section 2. Emergency response

table 1.Is it not missing a line referring to technical expertise and support as geotechnical and structural engineers?

4) section 3.3 causes

Concluding that a dead is because of the collapse of the house can not be so straightforward. Survival maybe be related to the length of time from the occurrence of the earthquake until the time of rescue from the rubble. Is there any data on that? number of hours or days that victims were trapped under the rubble?

4) section 3.4 analysis and section 4 lessons learnt

The critically or seriously injured people were quickly assisted, the medical response was adequate?were there infrastructures and medical personnel available? Analyzing the medical response can as well give hints to the lessons learnt.
* * *

---

## Author Comment (AC1) · 2 Jun 2020

Dear reviewer, Thank you for your comments for this paper. Your suggestions will help us to improve our research paper. I have the same feeling with you. Although the earthquake is not big, it is meaningful to analyze the cause of casualties. And I will consider increasing the contrast with other earthquakes to enrich my conclusion.

To the detail questions, our answers are as follows:

1) Abstract I'll take your advice. Replace the scale with a specific number, or add a specific number before the scale.

2) a)"Two or three earthquakes that are mentioned (two in section 1 and a third one at section 2)", I'm sorry, but I don't quite understand this suggestion. Can you help me

point it out in detail? I mentioned Wenchuan earthquake and Lushan earthquake in my article. Do you mean these two earthquakes?

b) As of 2014, the total population of Changning County is 464000, including 346000 permanent residents, 137000 urban residents, 95000 County residents and 5400 rural residents. I haven't found the data of the total number of houses in the disaster area yet, and I will continue to work hard to find it. 8889 billions yuan approximately equal to $1.25 billion, and I'll change it.

3) In the process of earthquake emergency response, technical expertise and support as geotechnical and structural engineers are indispensable. Experts providing technical support usually come from special working groups of departments such as Seismological Bureau and do not work as a separate department. But I will add this part of information in table 1.

4) Thank you for your advice. I will go further to find the field investigation report to determine the cause of casualties in more detail.

5) I will further consult the on-site investigation report and medical report to find out the details of the medical response. I will reflect my findings in the revision process of the article.

In the end, thank you for your help again, and this paper will benefit greatly from your thoughtful reviews.

---

## Referee Comment (RC2) · Anonymous Referee #2 · 13 Jun 2020

Dear author(s), I have read your article with interest. On this occasion, however, I have decided not to recommend the article for publication as it stands. It provides us some basic information about the earthquake and the emergency response, and the four types of the causes of the casualties are simply come from descriptive statistics of four different cases. However, this paper is more like a work report than a scientific research article, because there are only some brief description and summary about Sichuan Changning Ms 6.0 earthquake. Literature review, scientific methodology and logical analysis are essential for research paper, which are lacking in this paper. Regards.

---

## Author Comment (AC2) · 17 Jun 2020

Dear reviewer, Thank you for your comments for this paper. Your suggestions will help us to improve our research paper.

I have the same opinion with the other reviewer. Although this earthquake is not big, it is meaningful to analyze the cause of casualties. It is very important to analyze the cause of earthquake casualties and discuss the experience and lessons in China. All efforts to reduce casualties are worthy of recognition and attention. Authors obtained first-hand data by conducting field investigations of earthquakes, which contains the most detailed information about the name, sex, age, registered residence, ID number, cause of death, place of death of all the victims, and the information of the name, sex, age,

admission hospital, admission department, cause of injury and relevant information, diagnosis, home address, contact number of all the seriously injured person.

On the basis of sufficient field research and through the logical analysis of the causes of casualties, we give targeted improvement measures. This is also one of the scientific methodologies.

As for the lack of literature review, this paper was originally intended to focus on the analysis of the causes of casualties in this earthquake. Other reviewers also put forward the hope of enriching the data of other earthquakes, so I will adopt expert opinions to enrich my literature review and other earthquake comparison.

In the end, thank you for your help again, and this paper will benefit greatly from your thoughtful reviews.